# CO₂-Degassing Carbonate Conduits in Early Pleistocene Marine Clayey Deposits in Southwestern Umbria (Central Italy)

Angela Baldanza [1,*], Roberto Bizzarri [1], Chiara Boschi [2], Federico Famiani [3], Francesco Frondini [1], Marco Lezzerini [4], Steven Rowland [5] and Paul A. Sutton [5]

[1] Department of Physics and Geology, University of Perugia, via A. Pascoli, 06123 Perugia, Italy; roberto.bizzarri@libero.it (R.B.); francesco.frondini@unipg.it (F.F.)
[2] Institute of Geosciences and Earth Resources, National Research Council of Italy, 56127 Pisa, Italy; chiara.boschi@igg.cnr.it
[3] Parco e Museo Vulcanologico, Piazza Roma 1, 05010 Terni, Italy; federico.famiani@gmail.com
[4] Department of Earth Sciences, Università di Pisa, Via S. Maria 53, 56126 Pisa, Italy; marco.lezzerini@unipi.it
[5] School of Geography, Earth and Environmental Sciences, University of Plymouth, Drake Circus, Plymouth PL4 8AA, UK; srowland@plymouth.ac.uk (S.R.); p.a.sutton@plymouth.ac.uk (P.A.S.)
* Correspondence: angela.baldanza@unipg.it; Tel.: +39-75-5852632

**Abstract:** Early Pleistocene marine deposits in southwestern Umbria (Orvieto–Allerona area, Italy) recently revealed the presence of more than forty carbonate conduits distributed over 2 km along the Paglia riverbed. In order to investigate their origins, analyses of their mineralogy, $\delta^{18}$O and $\delta^{13}$C stable isotopes, and organic geochemistry were conducted. All the carbonate conduits are made of euhedral microcrystals of dolomite with subordinate quartz, plagioclases, and micas. The stable carbon and oxygen isotope values of the bulk concretionary carbonates range from −0.57 to +4.79‰ ($\delta^{13}$C) and from +1.58 to +4.07‰ ($\delta^{18}$O), respectively. The lack of organic geochemical biomarkers of anaerobic methane oxidation (AOM) and the very low values of extractable organic matter suggest a non-biological origin for the dolomite precipitation. The latter is probably related to the rise of volcanic carbon dioxide due to the incipient Vulsini magmatism recorded in Early Pleistocene marine deposits all around the study site. The spatial distribution of the structures indicates that the upward migration of the CO₂ was controlled by the fault system, while the vertical development of the conduits suggests that carbon dioxide degassing occurred, with multiple events. Carbon dioxide was probably stored in pockets within the clayey sediments until the pressure exceeded the eruptive threshold. These structures represent the first documentation of a volcanic carbon dioxide marine seepage event in the Umbria region.

**Keywords:** carbonate conduits; CO₂ emission; inorganic origin; volcanism; tectonics; early Pleistocene



## 1. Introduction

The discovery of mineralised structures attributable to sedimentary concretions emerging from the Early Pleistocene marine deposits in the Umbria region constitutes the first such report for this stratigraphic interval and is the research focus of the present paper.

Within the wide category of recorded sedimentary concretions, a peculiar group is represented by the so-called "carbonate conduits". These exhibit highly variable shapes (with pipe-, cylindrical-, and chimney-like morphologies) joined by the presence of internal conduits, which may be empty or filled with sediments [1,2].

In the summer of 2016, over forty large, mineralised structures of this kind were discovered in the area between the towns of Orvieto and Allerona, in southwestern Umbria (Italy), protruding from Early Pleistocene marine clay deposits. They were distributed over 2–4 km along the Paglia riverbed, exhumed from the present-day gravel river deposits either during the exceptional flood event of November 2012 or the subsequent restoration and

improvement work to restore the safety of the riverbed. At the moment, to our knowledge, they represent the first example of a carbon dioxide seepage event in the Umbria region.

These structures are morphologically very comparable to carbonate concretions produced by cold seep emissions, which are seafloor expressions of localised fluid flow in the marine environment [2–4].

Several fossil examples have been documented worldwide: in New Zealand [2,5–7], Colorado (USA) [8], Japan [9], the Outer Carpathians (Poland) [10], on the Montenegrin margin in the southern Adriatic Sea [11], and in the area of Pobiti Kamani (Varna, northeast Bulgaria) [12–14]. Several sites are also known in Italy: in the Northern Apennines [15]; at the Stirone River natural park [16–18]; on the Enza River bed near San Polo d'Enza, Reggio Emilia [19–21]; in the Badlands of Mt. San Pietro, Bologna [15]; in the Pietralunga, Sintria River Valley [16], and in the Cenozoic succession of the Tertiary Piedmont Basin [22,23].

These structures, containing chemosynthetic fossil taxa, have been documented in different tectonic settings and have often been associated with the venting of methane [2,3,5,8–10,15], although an active role for extensional faults can also be relevant for fluid circulation and mineralisation [2].

In contrast, in the present day and in the fossil records, only a few studies have reported mineralised conduits/chimneys related to carbon dioxide emissions (of organic and/or inorganic origin) [2,5,12,15,24–27].

The present work is focused on the Paglia river mineralised structures with the aim of highlighting (i) the mineralogical, petrological, and geochemical features, (ii) the origin of the fluids, and (iii) the mineralisation phenomena. Geological, volcanological, and paleoenvironmental implications are also discussed.

## 2. Geological Setting

The study area (Figure 1) pertains to the Miocene–Pleistocene South Valdichiana Basin [28], a NW–SE-oriented intermontane basin bounded by conjugated systems of extensional faults; it is part of the Neogene–Quaternary evolution of the Northern Apennines [29,30]. From the Pliocene onwards, the basin was filled with marine and continental deposits, and three main depositional cycles were recognised (Figure 2a; [28]):

- the "Pliocene" Cycle, widely documented in neighbouring basins [28,30,31], is suspected but not yet recognised in the study area. Although the deposits are age-equivalent to the FAA Fm (Formazione delle Argille Azzurre [28,31]), they are reported as Cycle I (Zanclean–Piacenzian hypothetic evolution of coastal areas) and offshore deposits (continuous/paraconformable distal marine sedimentation) [28].
- the Valdichiana Cycle (Gelasian–Calabrian), mainly consisting of the Chiani–Tevere unit and encompassing the higher lateral facies heterotopy and paleoenvironmental complexity of the area (Figure 2b), varying (from north to south and from coastal areas to the inner basin) from alluvial plains, deltas, river-fed beaches, rocky coasts, and shallow marine areas (shoreface to offshore transition), to open marine areas (~120–150 m deep [32,33]). This cycle is divided into three intervals by means of an integrated event stratigraphy (Figure 2a), and they are dated to Gelasian *p.p.* (Interval 1), Gelasian *p.p.*–Calabrian *p.p.* (Interval 2), and Calabrian *p.p.* (Interval 3) [28].
- the Middle–Late Pleistocene evolution (Cycle III [28]), which is mainly represented by Paleo–Trasimeno lacustrine deposits northwards and in the southern part of the basin by the volcanic and sedimentary units of the Vulsini Mts., Vico Mts., and Sabatini Mts. This latest cycle precedes the onset of the present-day valleys (Late Pleistocene–Holocene, Figure 2a).

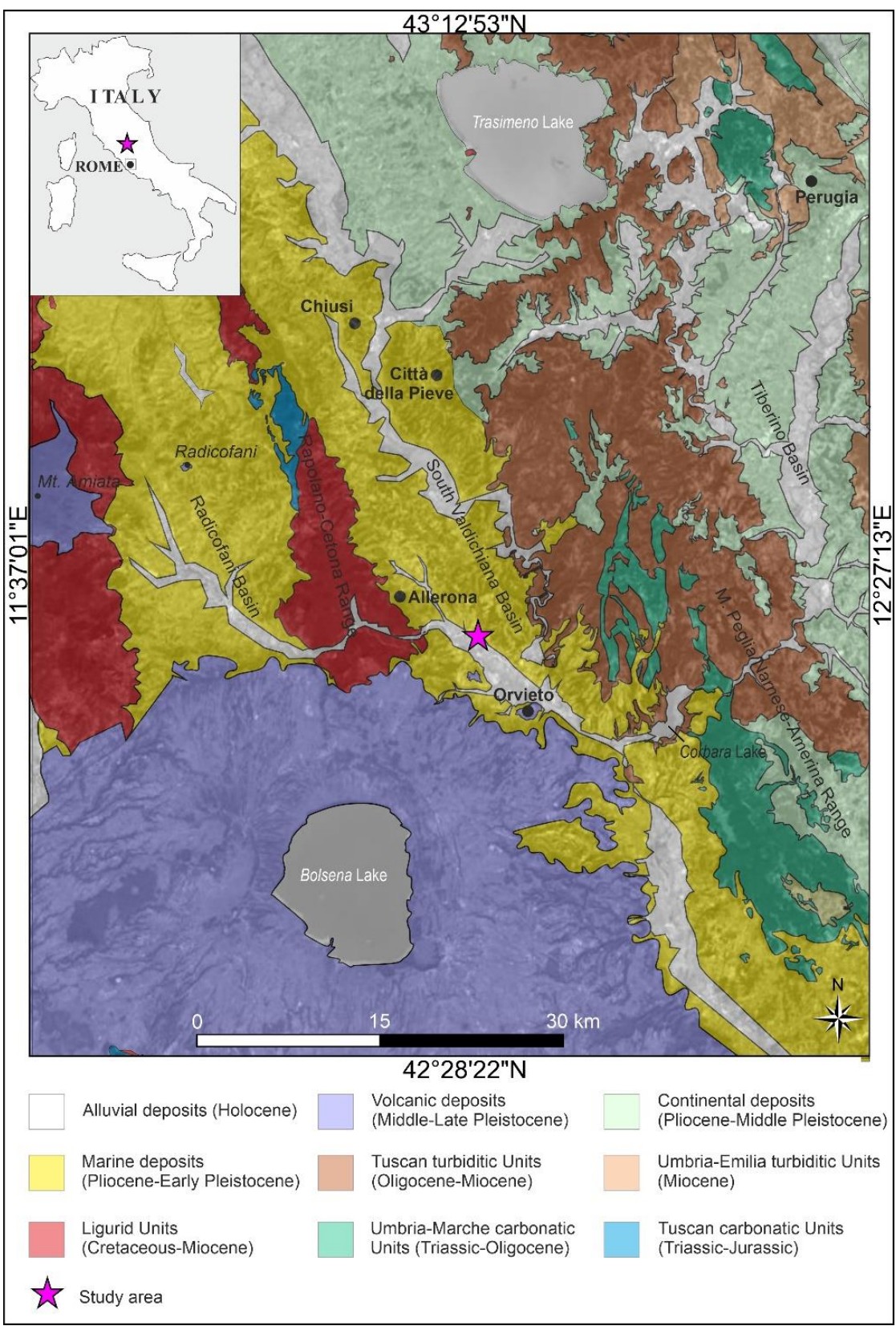

**Figure 1.** Simplified geological map of the study area (modified after [28]). In the inset, the position within Italy is shown.

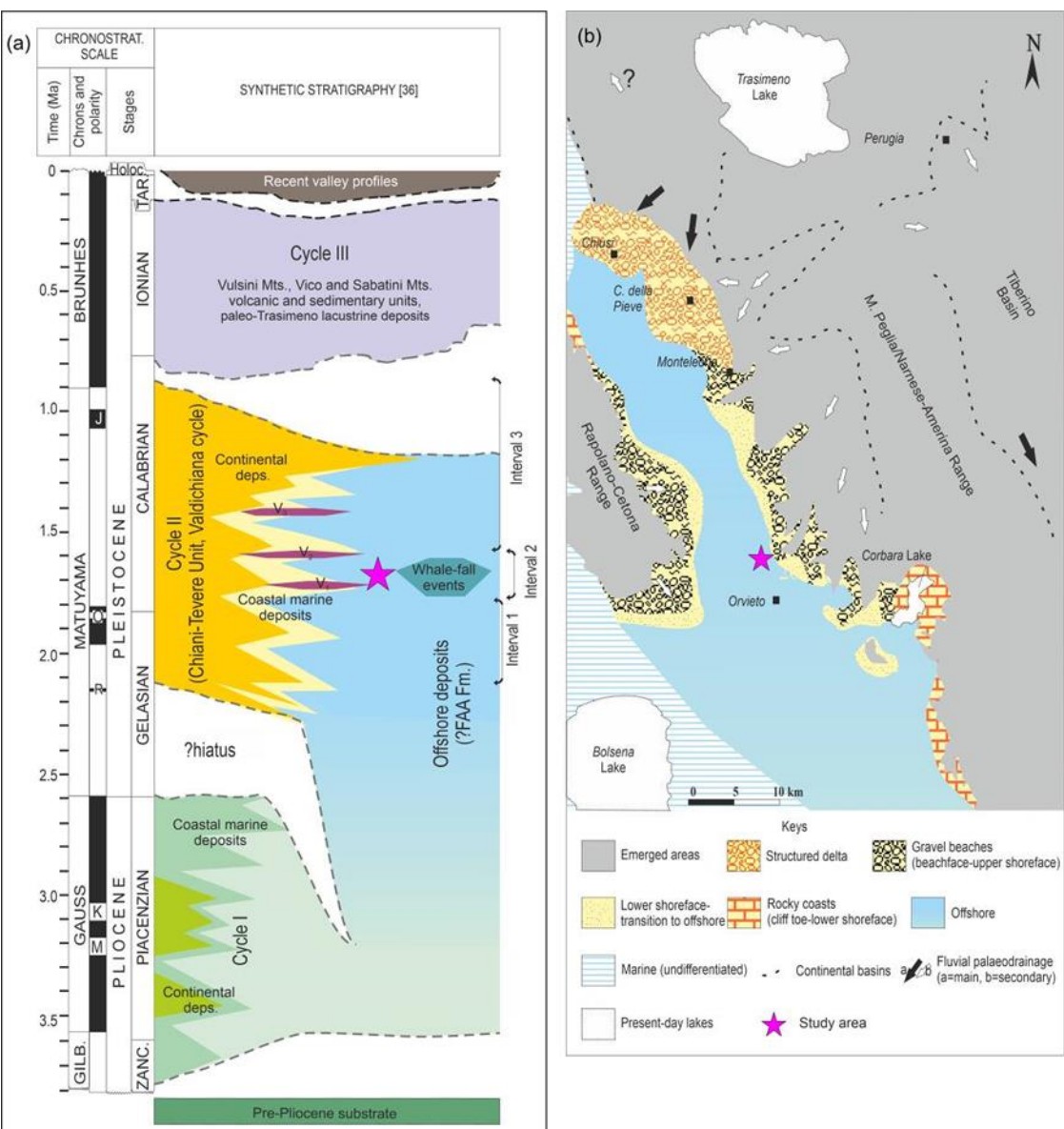

**Figure 2.** (**a**) Summarised stratigraphic scheme for the study area. (**b**) Early Pleistocene paleogeographic restoration. Modified after [28].

Volcanic deposits of the Vulsini Mts. Are mainly characteristic of the third depositional cycle, from ~750 ka to ~330 ka [34]. Nonetheless, older deposits, still referred to as resulting from Vulsini activity [35], were found in both marine and continental deposits of the Valdichiana Cycle [28] and dated to 1.7–1.4 Ma ($V_1$ to $V_3$ in Figure 2a). Together with whale-fall events, they highlight Interval 2 in the Valdichiana Cycle (1.75–1.59 Ma [28]). This narrow sea strait and its northern termination in the gulf of Chiusi-Città della Pieve [28,36] were frequented by cetaceans during the Early Pleistocene at least, and the presence of at least three whale-fall events (Figure 2a) has already been documented [33,37,38].

The Holocene evolution of the area in a fluvial/alluvial environment was mainly driven by the presence of two major rivers, the Paglia River and the Tiber River, and by related processes (e.g., vertical and areal erosion, valley calibration, river capture, lateral shift of channels, flood episodes, and slope processes [39,40]).

Even though it has been significantly modified by anthropogenic activities, the Paglia River follows a complex path, from the southern slopes of Mt. Amiata to its confluence in the Tiber River, near the village of Baschi (Figure 1). Particularly in the study area, the actual

riverbed cuts the Early Pleistocene marine clay sediments (Figure 3a [28,41,42]); at least one order of river terraces can be clearly recognised (Figure 3b,d). Although morphological evidence suggests a wider distribution (4 km or more in the NW–SE direction), mineralised structures were mainly found in a ~2 km narrow path along the Paglia River bends, a few kilometres north of the town of Orvieto (Figures 1 and 2). The mineralised structures (Figure 4) partly emerged from the Early Pleistocene deposits where the November 2012 exceptional flooding event removed part of the covering alluvial deposits (Figure 3b,c).

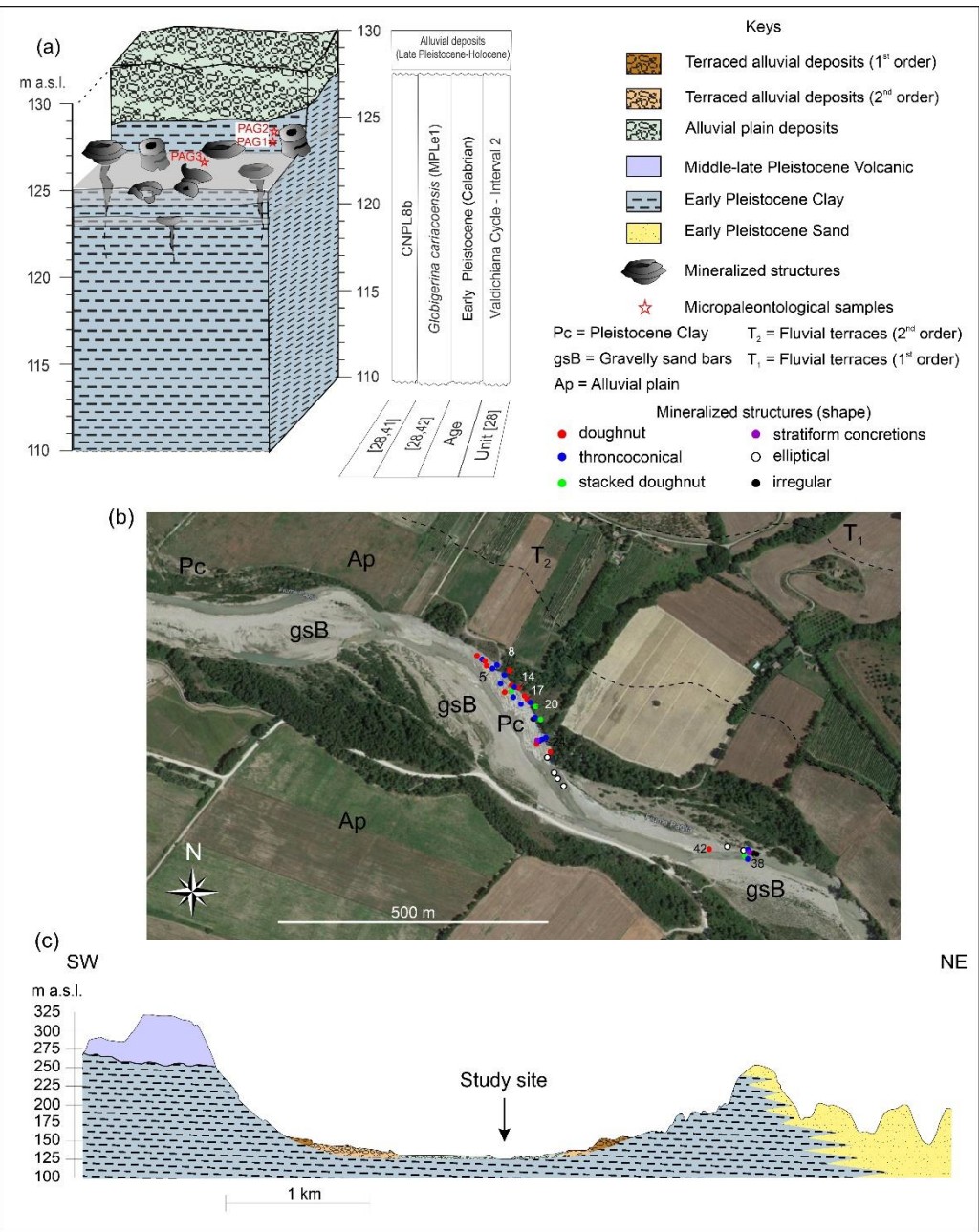

**Figure 3.** (**a**) Local section of deposits in the study site, stratigraphically referred to as the Early Pleistocene Valdichiana Cycle-interval 2 [28,41,42]. (**b**) Aerial view of the tract of the Paglia River and valley considered in this study, as it appeared in 2016, with the positions of the structures emerging from clay deposits on the riverbed. (**c**) The geological profile, which crosses the Paglia River Valley at the study site (vertical scale is exaggerated).

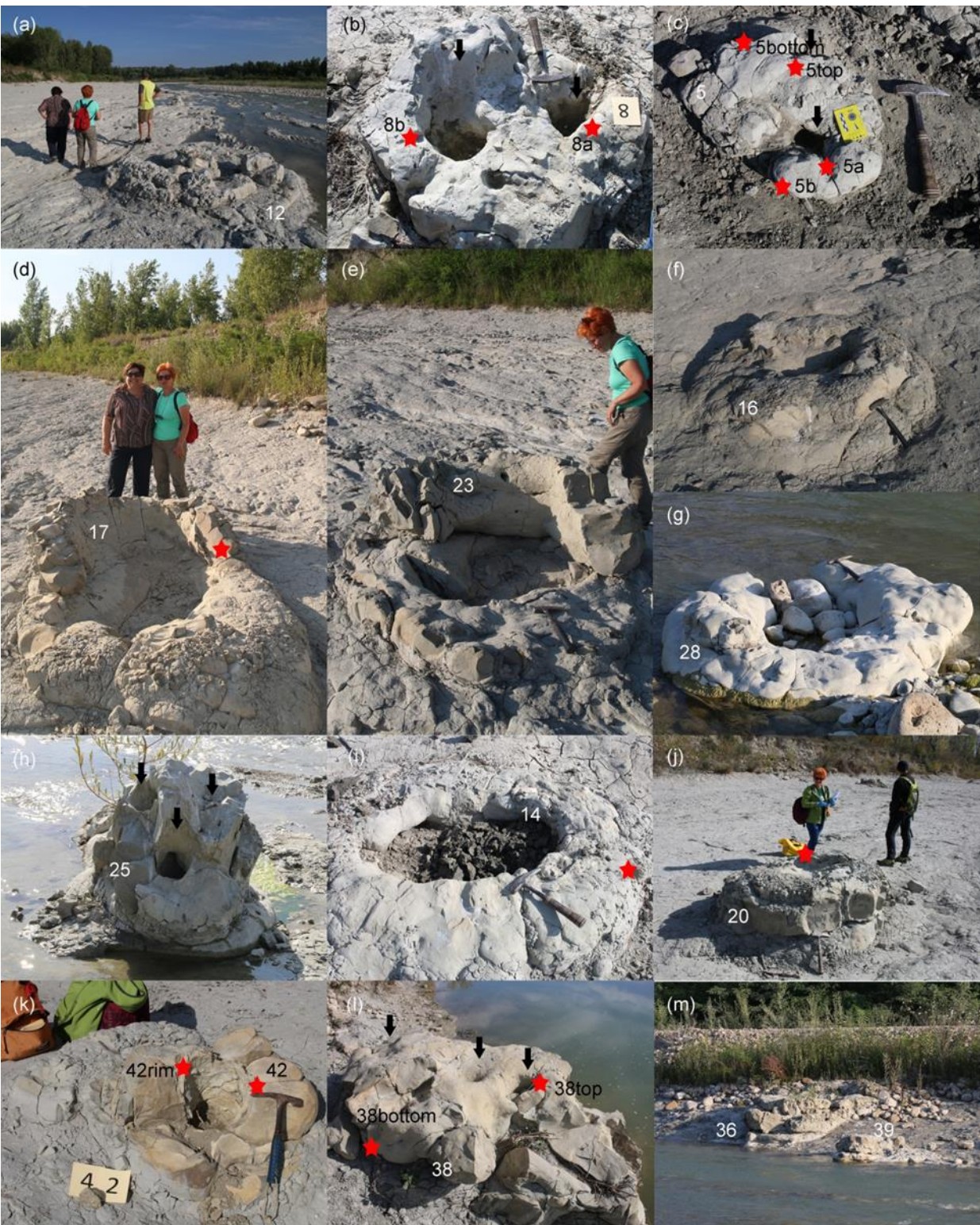

**Figure 4.** Carbonate conduit concretion morphologies. (**a**,**e**,**j**,**m**) Stacked doughnuts; (**b**,**c**,**h**) Tronco-conical shape: note that all possess double open mouths; (**d**,**f**,**g**,**i**,**k**) Ring doughnuts; (**l**) Stratiform concretion with small open mouths. Black arrows indicate the open mouths. Sizes and positions are given in Table 1. The white numbers refer to the list in Table 1, and the red stars indicate the sample positions.

## 3. Materials and Methods

Forty-two structures were described and measured in the field, and their positions were mapped by GPS (Table 1). On 8 different structures (i.e., the best-preserved and most complete), 14 samples were collected and processed for mineralogical, isotopic, and organic geochemical analyses. The sampling points for the 8 structures are detailed in Table 2 and Figure 4. Biostratigraphic and micropaleontological data were obtained for two samples (PAG 1 and PAG 2) of hosting clay sediments (collected just above structures #4 and #5) and 1 clay sample (PAG 3) collected outside of structure #14 (Figure 4i).

**Table 1.** Details of the morphology, sizes (diameters and heights), and GPS coordinates of the conduit concretions identified in the 2016 and 2017 surveys. Bold numbers indicate the analysed conduits.

| Conduit Concretions | Morphology | Diameters (Max, *min*) | Height | Coordinates | |
|---|---|---|---|---|---|
| 1 | doughnut | 60 cm | 10 cm | 42°45′27.99″ N | 12°4′55.19″ E |
| 2 | troncoconical | 30 cm | 20–25 cm | 42°45′27.69″ N | 12°4′55.72″ E |
| 3 | doughnut | M 68 cm; *m* 55 cm | 18 cm | 42°45′27.62″ N | 12°4′55.92″ E |
| 4 | doughnut | M 120 cm; *m* 62 cm | 30 cm | 42°45′27.27″ N | 12°4′56.11″ E |
| **5** | troncoconical | M 78 cm; *m* 60–76 | 30 cm | 42°45′27.04″ N | 12°4′56.65″ E |
| 6 | troncoconical | M 107 cm; *m* 92 cm | 40 cm | 42°45′27.35″ N | 12°4′57.01″ E |
| 7 | doughnut | M 360 cm; *m* 340 cm | 80 cm | 42°45′26.93″ N | 12°4′58.22″ E |
| **8** | troncoconical with 2 mouths | M 132 cm; *m* 125 cm | 72–73 cm | 42°45′26.56″ N | 12°4′57.77″ E |
| 9 | troncoconical | not measured/on the river flow | | 42°45′25.95″ N | 12°4′57.56″ E |
| 10 | doughnut | M 195 cm; *m* 165 cm | 85 cm | 42°45′25.89″ N | 12°4′58.50″ E |
| 11 | doughnut | not measured/on the river flow | not measured | 42°45′25.37″ N | 12°4′57.97″ E |
| 12 | stacked doughnut | M 380 cm; *m* 330 cm | not measured | 42°45′25.46″ N | 12°4′58.58″ E |
| 13 | troncoconical with two mouths | M 190 cm; *m* 90 cm | 75 cm | 42°45′25.75″ N | 12°4′58.73″ E |
| **14** | ring doughnut with large central mouth | M 185 cm; *m* 170 cm | 27 cm | 42°45′25.67″ N | 12°4′59.03″ E |
| 15 | troncoconical | not measured/on the river flow | | 42°45′25.07″ N | 12°4′58.79″ E |
| 16 | ring doughnut | M 190 cm; *m* 190 cm | 45 cm | 42°45′25.12″ N | 12°4′59.83″ E |
| **17** | ring doughnut | M 290 cm; *m* 190 cm | 70 cm | 42°45′24.81″ N | 12°4′59.89″ E |
| 18 | troncoconical with central mouth | M 80 cm; *m* 72 cm | 30 cm | 42°45′24.56″ N | 12°4′59.52″ E |
| 19 | troncoconical | M 88 cm; *m* 73 cm | 24 cm | 42°45′24.65″ N | 12°5′0.26″ E |
| **20** | stacked doughnut | M 200 cm; *m* 170 cm | 95 cm | 42°45′24.37″ N | 12°5′0.75″ E |
| 21 | troncoconical irregular | M 145 cm; *m* 125 cm | 35 | 42°45′23.62″ N | 12°5′0.80″ E |
| 22 | irregular | on the river flow | | 42°45′23.59″ N | 12°5′0.65″ E |
| 23 | stacked doughnut with central mouth and accessory small holes | M 175 cm +50; *m* 175 cm | 65 cm | 42°45′23.48″ N | 12°5′1.23″ E |
| **24** | troncoconical with two mouths | M 200 cm; *m* 188 cm | 50 (15 + 25 + 10) | 42°45′22.24″ N | 12°5′1.75″ E |
| 25 | troncoconical with two mouths | M 95 cm; *m* 25 cm | 50 cm | 42°45′22.19″ N | 12°5′1.46″ E |
| 26 | troncoconical with central mouth | M 115 cm; *m* 100 cm | 60 cm | 42°45′22.13″ N | 12°5′1.36″ E |
| 27 | doughnut | M 240 cm; *m* 180 cm | 40 cm | 42°45′21.87″ N | 12°5′1.15″ E |
| 28 | doughnut | M 190 cm; *m* 160 cm | 50 cm | 42°45′21.36″ N | 12°5′2.28″ E |
| 29 | stratiform concretions | on the river flow | | 42°45′22.10″ N | 12°5′1.09″ E |

**Table 1.** *Cont.*

| Conduit Concretions | Morphology | Diameters (Max, *min*) | Height | Coordinates | |
|---|---|---|---|---|---|
| 30 to 33 | elliptical | on the river flow | | ? | ? |
| 34 | elliptical spiralled | M 185 cm; *m* 160 cm | 15 cm | 42°45′15.85″ N | 12°5′16.05″ E |
| 35 | elliptical spiralled | M 200 cm; *m* 137 cm | 25–30 cm | 42°45′15.67″ N | 12°5′17.19″ E |
| 36 | very large stacked doughnut | not measured on the opposite riverbank | | 42°45′15.31″ N | 12°5′17.29″ E |
| 37 | crescent shape | on the river flow | | 42°45′15.63″ N | 12°5′17.54″ E |
| **38** | stratiform concretions | M 320 cm; *m* 280 cm | 135 cm | 42°45′15.51″ N | 12°5′17.66″ E |
| 39 | crescent shape | submerged | | 42°45′15.20″ N | 12°5′17.53″ E |
| 40 and 41 | irregular | submerged | | 42°45′15.52″ N | 12°5′17.98″ E |
| **42** | ring doughnut | M 60 cm; *m* 20 cm | 30 cm | 42°45′15.75″ N | 12°5′14.58″ E |

**Table 2.** Position of sampling points from selected structures and morphologies.

| Sample | Position | Structure Morphology |
|---|---|---|
| 5 top | periphery, upper surface | troncoconical |
| 5 bottom | periphery, ~40 cm below sample 5 top | |
| 5a | close to the mouth | |
| 5b | 10 cm from the mouth | |
| 8a | close to the mouth (a) | troncoconical, with two mouths |
| 8b | close to the mouth (b) | |
| 14 | periphery, upper ring | ring doughnut |
| 17 | periphery, upper ring | ring doughnut |
| 20 | periphery, upper ring | stacked doughnut |
| 24 | periphery, upper ring | stacked doughnut |
| 38 top | upper surface, close to the mouth | stratiform concretion |
| 38 bottom | 60 cm below sample 38 top | |
| 42 | periphery, external ring | ring doughnut |
| 42 rim | rim around the mouth | |

For each clay sample, 100 g was immersed in a solution of water and 5% hydrogen peroxide for 48 h. These were then rinsed and filtered with a Satylon retina (63 micron mesh) to remove particles smaller than 63 microns. The residues were dried in an oven at 70 °C. All the structure samples were processed for thin section preparation and analysed under a polarised light microscope (Zeiss, Department of Physics and Geology, University of Perugia, Perugia, Italy).

The sample mineralogical compositions were investigated using an X-ray powders diffractometer (XRPD) Bruker D2 Phaser 2nd Generation, under the following experimental conditions: X-ray tube, copper (Cu) type; wavelength 1.5418 Å at 30 kV and 10 mA; scanned range 2θ, varying from 5° to 65°, at a scan speed of 5°/min in steps of 0.05°.

Back-scattered electron (BSE) imaging and supporting geochemical analyses were conducted at the Center for Instrument Sharing at the University of Pisa (CISUP) using an FEI Quanta 450 FEG-SEM equipped with a standardless EDS system using a Bruker Quantax 400—Xflash detector.

Stable isotope analyses were performed using a Gas Bench II (Thermo Scientific, Waltham, MA, USA) coupled to a Delta XP IRMS (Finnigan) at the Institute of Geosciences and Earth Resources at the Italian National Research Council (IGG-CNR) in Pisa. Carbonate samples of ca.0.15 mg were dissolved in $H_3PO_4$ for 5 h at 70 °C. All the results were reported relative to VPDB and VSMOW international standards. Sample results were corrected using the international standard NBS-18 and a set of three internal standards, previously calibrated using the international standards NBS-18 and NBS-19 and by laboratory intercomparisons. Analytical uncertainty for both $\delta^{18}O$ and $\delta^{13}C$ measurements was ±0.1‰.

For the organic geochemical analyses, sediments were extracted as dried ground powders first with cyclohexane (CH) and then with dichloromethane (DCM). CH extracts were examined directly for organic geochemical biomarker hydrocarbons of the anaerobic oxidation of methane (AOM) by gas chromatography (GC) methods, including GC, high-temperature GC (HTGC), GC–mass spectrometry (GC–MS) and HTGC–MS. DCM extracts were examined similarly for polar biomarkers of AOM, but this examination occurred after derivatisation with BSTFA (cf [5]). Finally, residues after CH and DCM extraction were methylated, and the methyl esters were also examined by the above GC methods. Detailed descriptions are provided in the Supplementary Material [43–47].

## 4. Results

### 4.1. Conduit Concretion Morphology and Spatial Distribution

The conduit concretions ("all concretions possessing conduits", sensu [2]) along the Paglia riverbed, which are also visible in the aerial photos from 2015 to 2019 (Figures 3b and 4), were geolocated and measured directly in the field and the data are reported in Table 1.

Minimum/maximum diameters were slightly variable (Table 1), from 20 cm to more than 300 cm, with no clear relation between dimensions and morphologies.

The main common morphologies (following the nomenclature of [1,2]) were doughnuts (with two different shapes: ring doughnut and stacked doughnut) and troncoconicals (Figure 4). Locally stratiform concretions (sensu [2]) also occurred. The ring doughnuts (Figure 4d,f,g,i,k) were characterised by a single central mouth variable in size from 30 cm to over 150 cm. The larger elliptic specimen (Figure 4d) showed small circular openings (variable in size from 1 to 3 cm) placed in the inner portion of the wall. The stacked doughnuts (Figure 4a,e,j,m) were the largest and consisted of three or four perfectly stacked doughnuts, but these often showed partially preserved outlines. Additionally, in this case, small circular openings (Figure 4e) were visible in the inner portion of the walls. The troncoconical morphology was most common (Figure 4b,c,h) and possessed one or two open conduits (mouths), generally of similar diameters. The stratiform concretions (Figure 4l) were rarer and mainly emerged in the middle of the watercourse; they were massive and elongate and supported at least 10 openings with diameters variable from 1 cm to 25 cm. This morphology, according to [2], could represent the lateral coalescence of several conduits.

### 4.2. Mineralogical and Petrographic Features

Macroscopic observations in the field revealed that the conduit concretions were mainly light grey. Only two (structures 17 and 42, Figure 4d,k) were hazelnut in colour. The mouths of conduits were covered with a thin (from 1 mm to 2 mm), striated, smooth, light-yellow finish, which mineralogical analysis recognised as calcite.

Microscopic examinations of thin sections and XRPD analyses showed that the samples were composed of two types of matrixes: the first (structures 5 and 14) was represented by micropeloids (aggregates of about 1 mm or larger), composed of microcrystalline dolomite and rare micrite (Figure 5a–c). The second (structures 8, 17, 20, 24, 38, and 42) was a fine homogeneous matrix of dolomite microcrystals with common pyrite crystal aggregates (Figures 5d and 6a) and randomly distributed filamentous structures with diameters up to 10 µm, and lengths variable from 40 to 200 µm (Figure 5c,d).

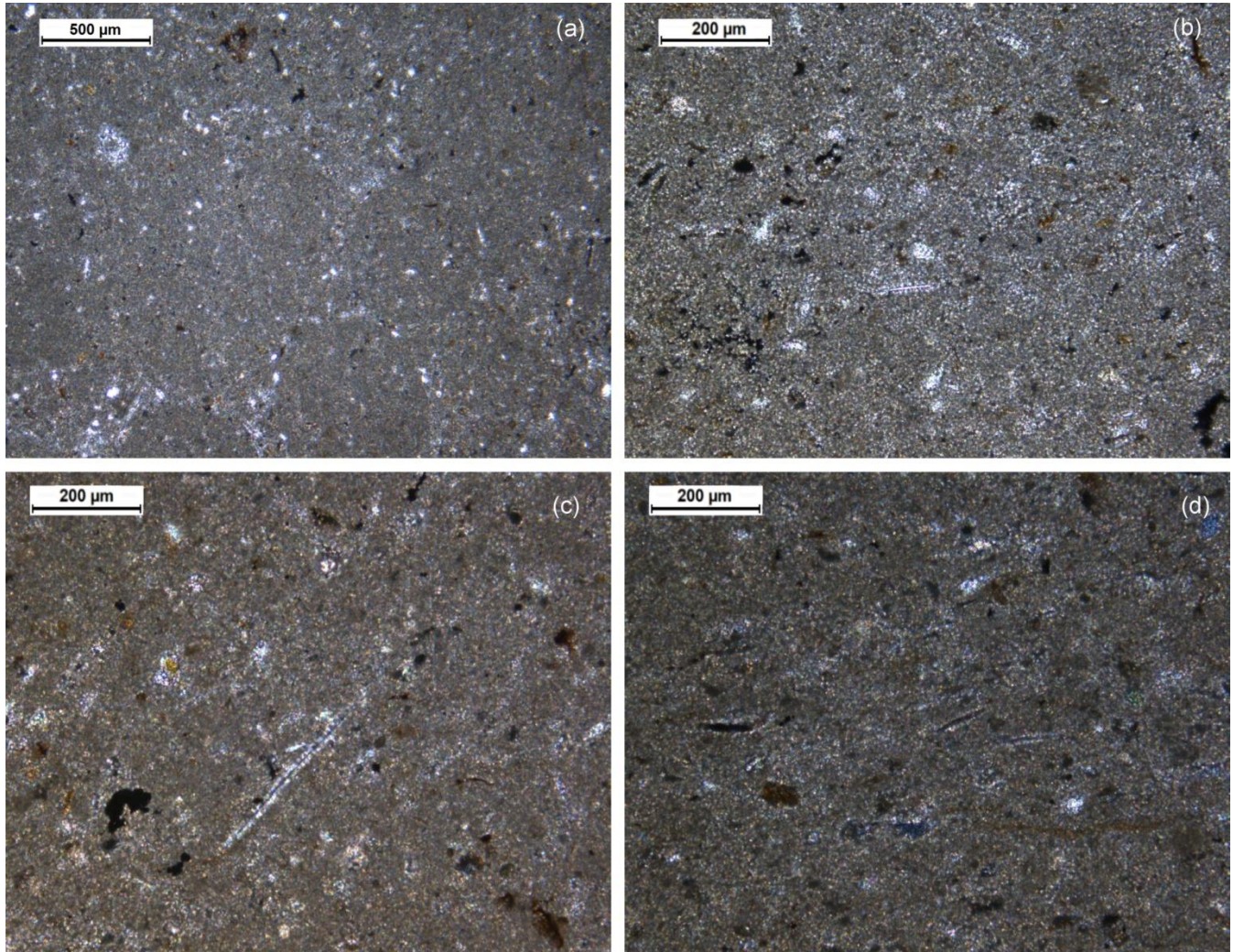

**Figure 5.** Optical polarised light microscope microphotographs (**a**–**d**): (**a**) structure 5; (**b**) structure 14; (**c**) structure 20; (**d**) structure 42, rim.

The detrital fraction always includes quartz in association with traces of feldspars, micas, kaolinite, apatite, and pyrite (Table 3).

SEM–EDS analyses revealed that the filamentous structures were represented by clay minerals completely enveloped by euhedral dolomite microcrystals (Figure 6c,f). The clay minerals commonly occurred, as did very fine silt-sized grains of quartz and plagioclase, as well as a few foraminifer tests filled by pyrite (Figure 5a). The very small size of the dolomite (from 2 to 25 µm) and the low porosity are indicative of a very high precipitation rate.

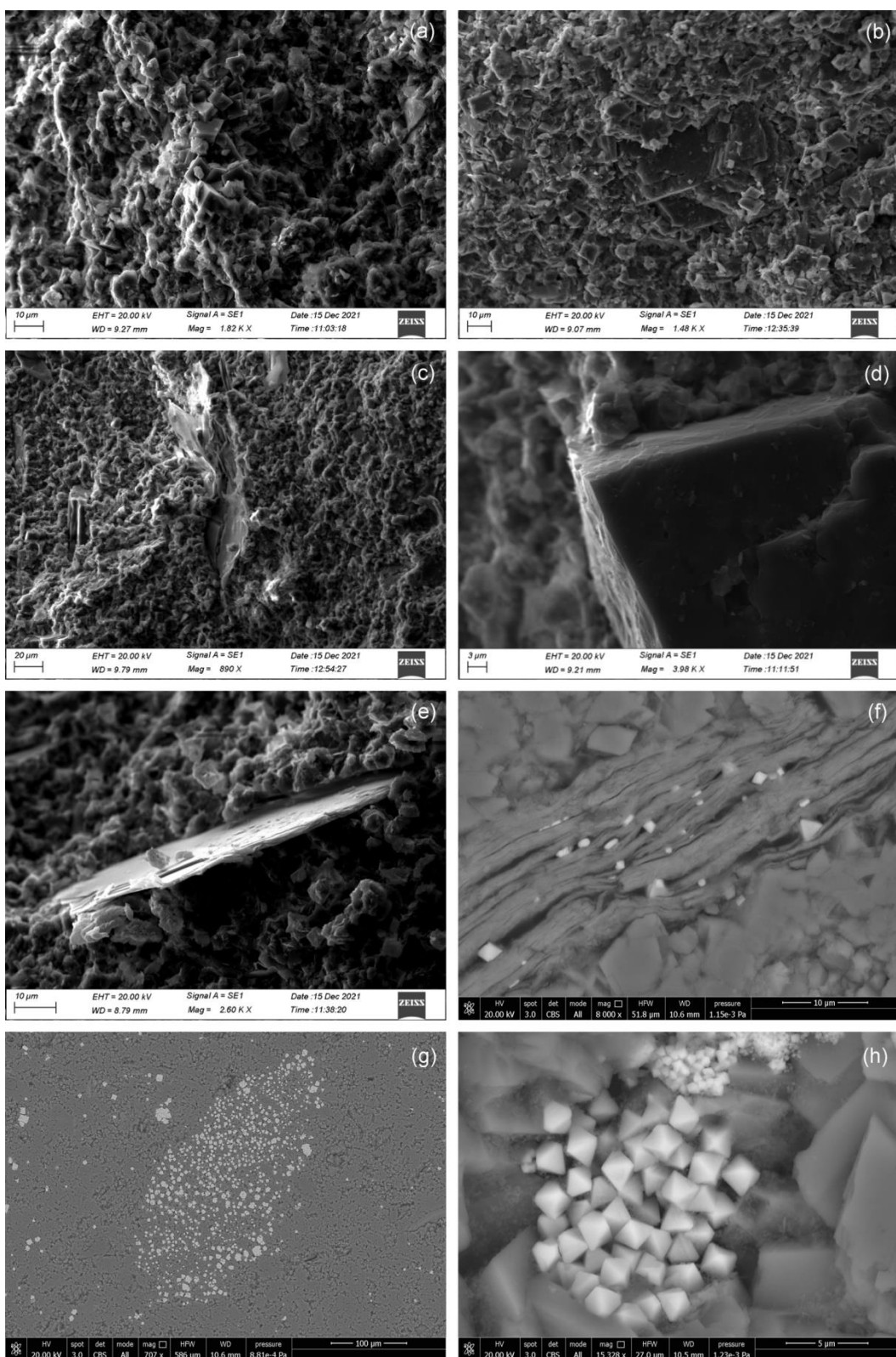

**Figure 6.** Scanning electronic microscope microphotos showing dolomite and phyllosilicates: (**a**) structure 5a; (**b**) structure 42, rim; (**c**) structure 20; (**d**) structure 5a; (**e**) structure 5b; (**f**–**h**) Back-scattered electron (BSE) images of structure 17 showing (**f**) detail of phyllosilicates; (**g**,**h**) details of pyrite microcrystals embedded into dolomite microcrystals.

**Table 3.** Mineralogical composition determined by XRPD and SEM/EDS analyses, values of stable isotopes ($^{13}$C, $^{18}$O), and Z value of the eight conduit concretions and minerals detected. Z is calculated according to Formula (1).

| Samples | Dolomite | Calcite | Quartz | Plagioclase | Mica | Kaolinite | Apatite | Pyrite | $\delta^{13}$C | $\delta^{18}$O | Z |
|---|---|---|---|---|---|---|---|---|---|---|---|
| 5 top | XXX | X | X | tr | tr | tr | tr | tr | 0.68 | 1.58 | 129.47 |
| 5 bottom | XXX | tr | X | tr | tr | tr | tr | tr | 0.77 | 2.45 | 130.09 |
| 5a | XXX | | X | tr | tr | tr | tr | tr | 0.36 | 3.13 | 129.59 |
| 5b | XXX | | X | tr | tr | tr | tr | tr | −0.41 | 3.44 | 128.17 |
| 17 | XXX | | X | tr | tr | tr | tr | tr | −0.57 | 4.07 | 128.10 |
| 8a | XXX | | X | tr | tr | tr | tr | tr | 1.45 | 2.89 | 131.70 |
| 8b | XXX | | X | tr | tr | tr | tr | tr | 2.30 | 3.13 | 133.56 |
| 14 | XXX | | X | tr | tr | tr | tr | tr | 4.45 | 3.27 | 138.04 |
| 20 | XXX | | X | tr | tr | tr | tr | tr | 4.79 | 3.66 | 138.93 |
| 24 | XXX | | X | tr | tr | tr | tr | tr | 4.33 | 3.64 | 137.98 |
| 38 top | XXX | | X | tr | tr | tr | tr | tr | 3.60 | 3.39 | 136.36 |
| 38 bottom | XXX | | X | tr | tr | tr | tr | tr | 3.23 | 3.23 | 135.52 |
| 42 | XXX | | X | tr | tr | tr | tr | tr | 2.46 | 3.67 | 134.17 |
| 42 rim | XXX | | X | tr | tr | tr | tr | tr | 4.05 | 3.01 | 137.09 |

XXX = very abundant, X = present, tr = trace.

### 4.3. Organic Geochemical Biomarkers

Organic geochemical biomarkers of anaerobic methane oxidation (AOM) are well-known (e.g., [5,48]) and include the hydrocarbons crocetane [49] and pentamethylicosane (formerly known as pentamethyleicosane [50]), as well as the ether archaeol [51], all of which are likely biological products of archaea and related microbes [52]. The $^{13}$C isotopic values of these compounds are also often very distinctive (typically −150‰ [48]). Unfortunately, none of these compounds was detectable in any of the structures examined, despite the use of appropriate methods.

Indeed, the amounts of extractable organic matter in the samples were very low (typically 0.1–0.5 mg per 5 g sample, with 0.1 mg in the blank). The hydrocarbons and other more polar compounds that were detected were mainly atypical of indigenous material. For example, sterane and triterpane biomarkers showed distributions mainly typical of mature petroleum, unexpected in Pleistocene sediments (c.f., the immature distributions found by [5] in Miocene deposits in New Zealand). Although other commonly used biomarker parameter estimates (e.g., Carbon Preference Index, CPI; Odd over Even Preference, OEP; and Average Chain Length, ACL of *n*-alkanes; Tables S1 and S2) varied according to the method of calculation; nonetheless, CPI values around 1.0 were typical (Table S2). These also indicated the presence of thermally mature hydrocarbons, as did the presence of unresolved complex mixtures of hydrocarbons [53]. Squalene was also present in most hydrocarbon extracts but is not diagnostic and may be an artefact; phthalate esters, which are common plasticisers, were also identified in the more polar fractions [54].

### 4.4. Stable Isotopes

The isotopic compositions of carbon and oxygen in the carbonates may indicate the conditions under which they formed. The $^{13}$C ratio values indicate the carbon sources that may have participated during carbonate formation (biogenic, abiogenic, or thermogenic methane, or their mixtures). The $^{18}$O ratio values allow for the determination of the pore water temperatures in which carbonate minerals precipitate. In addition, C–O stable isotope values could be useful for evaluations of the paleo-salinity of the water and can be inferred from the Z value [55], calculated according to the formula:

$$Z = 2.048(\delta^{13}C + 50) + 0.498(\delta^{18}O + 50) \tag{1}$$

If Z < 120, this indicates a freshwater environment; if Z > 120, this indicates a marine environment or the participation of seawater during precipitation.

$\delta^{13}$C values of the bulk concretionary carbonate (dolomite) in this study ranged from −0.57 to +4.79‰ PDB (average + 2.22‰). $\delta^{18}$O values ranged from +1.58 to +4.07‰ PDB (average +3.23‰; Table 3). The Z values of the samples were higher than 120, which indicates the influence of seawater in providing the $Mg^{2+}$ source for the formation of the dolomite.

There is no clear trend in the isotope values from the centre or the rim of the dolomite-rich conduits, suggesting an erratic precipitation.

*4.5. Hosting Sediments*

The clay deposits containing the conduit concretion field were dated to Early Pleistocene and associated with Interval II in the Valdichiana Cycle [28].

The grey clay marine deposits, occurring in a small outcrop (50 cm in thickness) above the conduit concretion surfaces (samples PAG 1 and PAG 2) and a few centimetres next to structure #14 (sample PAG3), were treated to obtain washed residues for micropaleontological analyses. The washed residues (~1 g) of samples PAG 1 and 2 contained moderate to rich assemblages of microfossils represented by benthic and planktonic foraminifers and very few fragments of bivalves or small carbonised wood fragments. The planktonic foraminifers were represented by the common *Orbulina universa* and the rarer *Globorotalia inflata*, *Globigerina cariacoensis*, and *Turborotalita quinqueloba* in assemblage with *Globigerinoides ruber* and *Globigerinoides sacculifer*. The benthic foraminifers were represented by both epifaunal and shallow infaunal species. The detritivores *Lenticulina calcar* and *Siphotextularia concava* and the herbivorous *Cibicidoides lobatulus* were abundant among the epifauna, and *Nonionella turgida* and *Vaginulina striatissima* were abundant among the shallow infauna. *Hyalinea balthica* was rare but always present in assemblages. In particular, the occurrence of *H. balthica* indicates a bottom temperature ranging between 4 and 12 °C [56]. The occurrence of *L. calcar* and *H. balthica*, both inhabitants of cold water, indicates that the sea floor was cool and the sediment rich in organic matter (or phytodetritus). This last factor is evidenced by the abundance of the species *S. concava* and *N. turgida*, which prefer sediments rich in organic matter but do not tolerate high organic fluxes [37]. Moreover, the occurrence of shallow infaunal benthic species (*V. striatissima*, *N. turgida*, and *H. balthica*) indicates normal oxygenation at the water–sediment interface, and the abundance of epiphytal and herbivorous *C. lobatulus* suggests the existence of a sea grass prairie.

The assemblage of sample PAG3, collected next to structure 14 (stratigraphically below sample PAG1), was quite different, with a decrease in planktonic species, represented only by the common *O. universa* and the very rare *Globigerina bulloides*, *G. cariacoensis*, and *G. ruber*, all affected by pyritisation. The benthos were enriched with the entrance of shallow infaunal species such as *Marginulina costata* and *Bigenerina nodosaria*; that added up to very abundant *L. calcar* and *N. turgida*.

**5. Discussion**

*5.1. Origin of the Fluids*

Dolomite commonly makes up only a small percentage of seep carbonate lithologies, but sometimes—as in the present work—it is the dominant mineralogy. Its formation and distribution are often ambiguous. Most authors report the occurrence of seep-related dolomite as strictly related to low-temperature microbially mediated reactions in the shallow subsea floor ([57] and references therein).

In particular, in methane seeps, the anaerobic consumption of seawater sulphate by consortia of archaea and sulphate-reducing bacteria causes a depletion in $MgSO_4$ and is accompanied by an increase in the concentration of free Mg ions and an increase in pore water alkalinity and eventual $HCO_3^-$ supersaturation, thereby favouring the preferential formation of dolomite [2].

However, most microbially mediated low-temperature primary dolomite formation that directly precipitates is characterised by organic biomarkers and negative $\delta^{13}$C values, as reported in Figures 7 and 8.

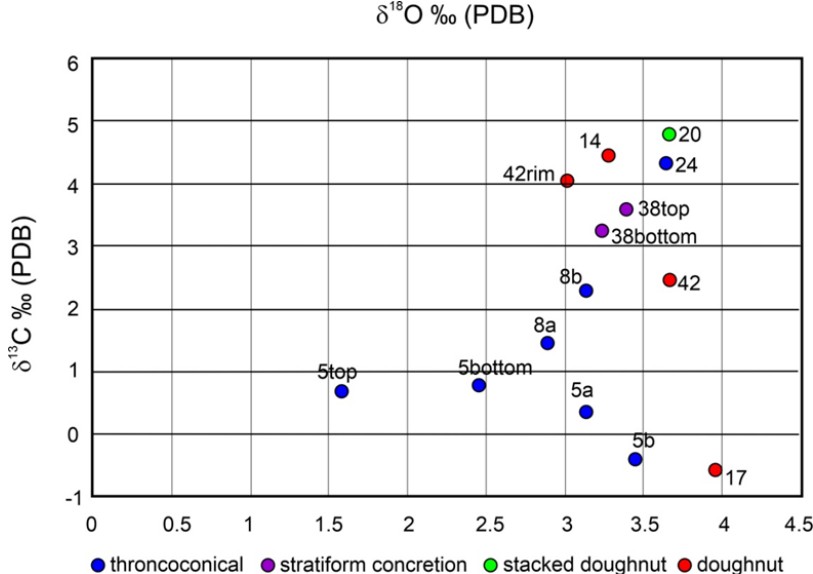

**Figure 7.** Stable carbon ($\delta^{13}$C) and oxygen ($\delta^{18}$O) isotope ratios plot for selected conduit concretions from the Paglia riverbed. Numbers and shapes used are the same as in Table 1. Isotopic values are reported in Table 3 ($\delta^{13}$C and $\delta^{18}$O values relative to the different morphologies of carbonate concretions).

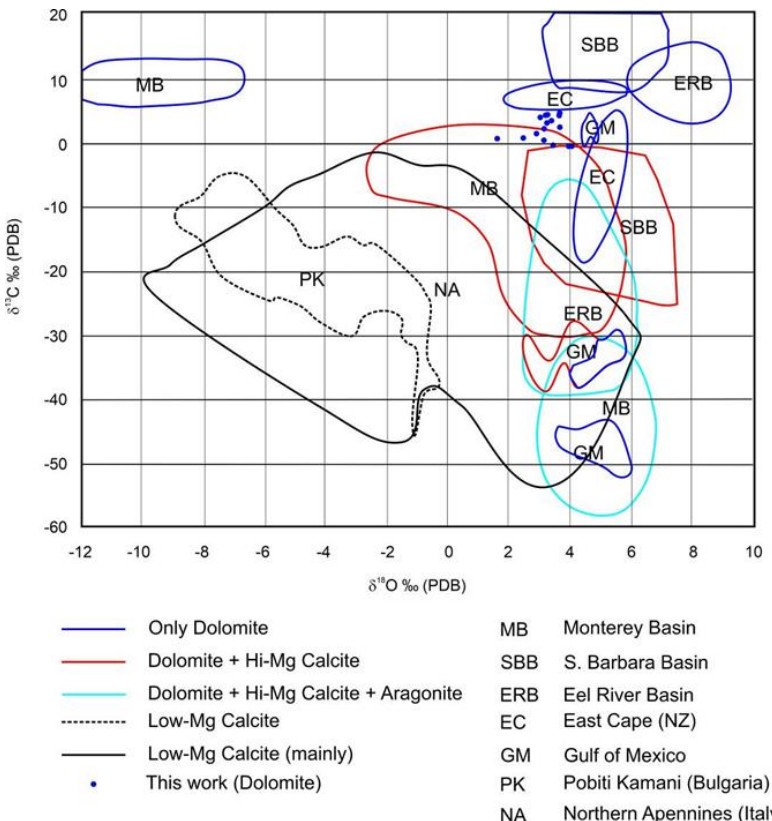

**Figure 8.** Plot of carbonate conduit mineralogy and stable carbon ($\delta^{13}$C) and oxygen ($\delta^{18}$O) isotope values from different world examples. Data are from [2,12,15,57,58].

Our studies indicate a mineralogical composition of the conduits dominated by microcrystalline euhedral dolomite, with minor plagioclase, mica, kaolinite, quartz, very few foraminifers, and with no, or rare, organic material.

This evidence, together with the positive values of $\delta^{13}C$ and the absence of organic geochemical biomarkers of anaerobic methane oxidation (AOM), may suggest an unusual, inorganic origin for the seep conduits, unrelated to microbially mediated reactions.

The positive $\delta^{13}C$ values can be compared with those found in some East Cape dolomite concretions ($\delta^{13}C$ +6 to +9‰ vs. PDB) [2] (Figure 8) and interpreted as sourced from reduced $CO_2$ as a by-product of methanogenesis and derived from the microbial production of methane during a shallow burial (10–100 m) of the sediment pile. The East Cape concretion-related biomarkers confirmed that AOM by methanotrophic archaea and sulphate-reducing bacteria occurred in the sediments [5], even though the $\delta^{13}C$ values of the carbonate cements are dominantly positive [2].

However, the absence of organic geochemical biomarkers of anaerobic methane oxidation (AOM) in our dolomite concretions seems to exclude this possibility. The other remaining possible origin explaining the observed $\delta^{13}C$ values is the direct emission of hydrothermal/volcanic carbon dioxide.

In the study area, the inorganic origin of $CO_2$-rich fluids is probably related to a process of $CO_2$ diffusion degassing associated with volcanic activity starting from the Early Pleistocene (about 1.7 Ma) [59].

The $\delta^{13}C$ values of carbonates, ranging from −0.57 to +4.79‰ vs. PDB, are consistent with a magmatic origin of the gas phase. The fluids released by Italian volcanoes and geothermal regions are characterised by $\delta^{13}C$ values of the original gas phase (before dilution or fractionation) between −3 and −1‰ vs. PDB (e.g., [60–62]). Similar values are also computed by [63], to explain the isotopic evolution of Vesuvio groundwaters, while the $\delta^{13}C$ values of the Campanian submarine gas emissions range from −1.8 to −2.3‰ vs. PDB.

Starting from this range of $\delta^{13}C$ values for the deep gas phase, we suggest that the more positive carbon isotopic composition of our solid samples is explained by the fractionation that occurred during the degassing and subsequent carbonate precipitation phases at low temperature, which can lead to a positive isotopic shift of some delta units [63]. The accurate value of the isotopic shift cannot be calculated for our samples because it depends on the pressure at which degassing occurred and the possibility of a multiple-step degassing process. As an example only, [63] calculated an isotopic shift of about +4 delta units from the continuous degassing of a saturated 24-bar $CO_2$ solution brought to a pressure of 1 bar.

Oxygen isotope values recorded in the carbonate samples relate mainly to the fluid composition at the time of carbonate precipitation and the temperature of precipitation.

Considering that the $\delta^{18}O$ values of our samples range between 1.58 and 4.07‰ vs. PDB (Table 3 and Figure 7) and assuming the present seawater $\delta^{18}O$ (0‰ SMOW), the calculated temperature of precipitation, following the fractionation factor of [64], ranges from 20 to 30 °C.

In contrast, the benthic epifaunal and shallow infaunal foraminifers present in the hosting sediments indicate that the temperature at the bottom was low, specifically, in the range between 4 and 12 °C, marking permanent cold bottom water conditions and a cold overlying water column.

In order to explain the temperature of dolomite precipitation inferred from the oxygen isotope values, we propose that warm, $CO_2$-rich fluids rose through the carbonate conduits and mixed with this cold seawater.

### 5.2. Formation of Dolomite Conduits

The large dolomite-rich doughnut concretions were associated by [2] with intermittent fluid ascent ($CO_2$ in the present case), where fluid periodically becomes trapped by relatively more impervious (perhaps clay-enriched) stratigraphic layers. The authors of [6] suggested that seep-related tubular (conduit) concretion formation occurred in the subsur-

face, from cement precipitation, starting at the outer rims of concretions and continuing toward the centre of their conduits.

In our case, the centre of each structure is very open (as shown in Figure 4), unlike those reported by [2,6], which had completely filled central areas (mouths).

The free and well-formed mouths, covered with a thin (from 1 mm to 2 mm) calcitic patina with longitudinal striae (Figure 4b,i,h,l), underline the rise of a vertical flow under pressure (presumably a mixture of gas and liquid).

The calcite crystals were also found near the small (centimetre) mouths distributed along the internal rims of the doughnut structures (Figure 4e,l).

### 5.3. $CO_2$ Flux and Dolomite Stability

In order to explain the ubiquitous occurrence of dolomite in the carbonate conduits, it is necessary to investigate the influence of $CO_2$ on the stability of the carbonate mineral species. The presence in the study area of an Early Pleistocene volcanic degassing zone is suggested by the occurrence of distal pyroclastic-fallout materials in the marine clay sediments [28,34,59]. These deposits, with ages between 1.75 and 1.2 Ma, have a local origin, probably from a small eruptive centre near Orvieto. Considering the composition of the volcanic products, it is likely that the degassed phase was very similar to the gas emitted in the present day from Latera and Torre Alfina, located a few kilometres south of the study area, and by the $CO_2$-rich springs of Parrano and Monte Rubiaglio (Figure 9, [65–68]).

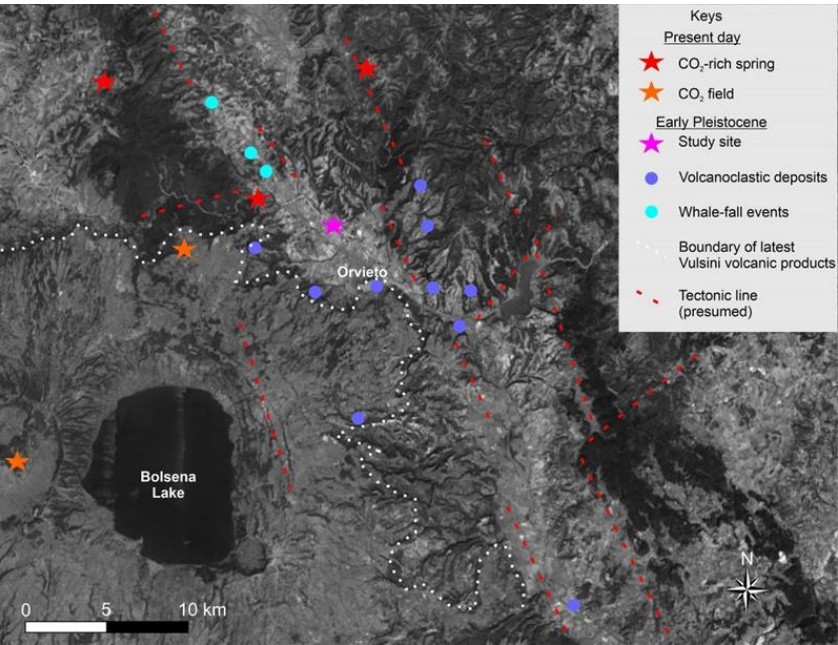

**Figure 9.** Aerial distribution of Early Pleistocene volcanoclastic deposits, whale-fall events (WFE), and present-day emergences of $CO_2$; the boundary of the latest (Late Pleistocene) Vulsini volcanic products and the main tectonic lines are also shown. Data are from [28,34,59,65].

If the degassing process occurred in a relatively shallow marine environment, the volcanic $CO_2$ seepage could have produced high values of $CO_2$ partial pressure ($pCO_2$) and the acidification of seawater, with the consequent undersaturation in calcite and aragonite, similar to that which is currently observed in present-day submarine volcanic hydrothermal fields [69].

In order to describe the possible interactions of the gas phase with seawater and with carbonate minerals, a theoretical water–gas–rock interaction model was constructed, starting with 1 kg of average seawater with a salinity of 35 ppt [70] and increasing the $CO_2$ concentration from 0.02% to 0.5% in 96 steps at 10 °C. At each step, the saturation indexes (SI) of calcite, aragonite, and dolomite, along with pH and $pCO_2$, were computed with

the PHREEQC code [71,72] using the LLNL (Lawrence Livermore National Laboratory) thermodynamic database [73].

The results of the calculations (Figure 10) show that for $pCO_2$ values normally found in shallow seawater, the solution is supersaturated in calcite, aragonite, and dolomite. In these conditions, even if dolomite had an SI higher than calcite, the formation of calcite would predominate over dolomite because of the relatively slow rates at which dolomite forms [67]. For increasing values of $pCO_2$, the pH of the solution and the SI values of all the carbonate species decrease. At $pCO_2$ values higher than $10^{-2.75}$ bar, the solution is undersaturated in calcite and aragonite but is still oversaturated in dolomite. In these conditions, the only carbonate mineral that can precipitate is dolomite. For $pCO_2$ values higher than $10^{-2.35}$, dolomite is also undersaturated, and no carbonate mineral is allowed to precipitate.

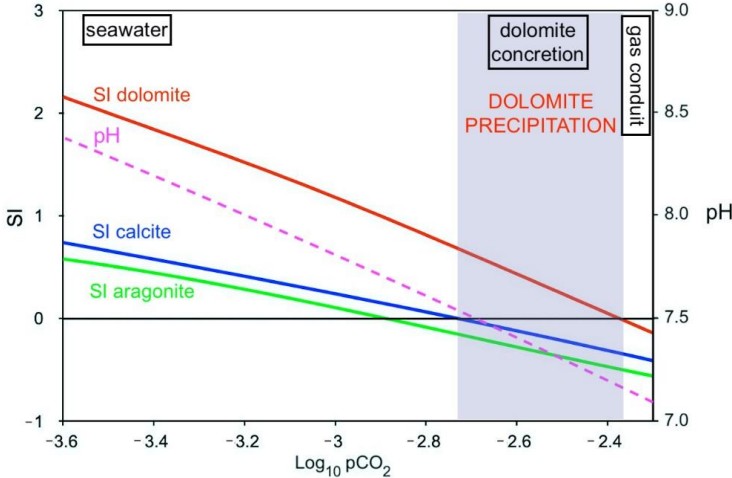

**Figure 10.** Variations of dolomite, calcite, and aragonite saturation indexes (SI) and pH, as a function of $Log_{10}\ pCO_2$. SI values higher than 0 indicate supersaturation, and values lower than 0 indicate undersaturation.

According to the proposed geochemical model, the presence of $CO_2$ seeps could produce, at the sediment–water interface, a narrow region around the gas conduit characterised by $7.5 > pH > 7.1$ where dolomite precipitation is possible. For lower pH values, corresponding to $pCO_2 > 10^{-2.3}$, dolomite precipitation is thermodynamically not permitted, keeping the gas conduit open and allowing the progressive growth of the chimneys.

*5.4. Effects of $CO_2$ Flow in the Marine Environment and Inhabitants*

Recently, the geochemical aspects of shallow marine $CO_2$-rich seeps worldwide have been reviewed [69], focusing on both gas composition and water chemistry. The authors of [69] describe the geochemical effects of volcanic $CO_2$ seepage on the overlying seawater column. In the $CO_2$ seep sites off Vulcano Island (Sicily), in areas of intense bubbling, extremely high levels of $pCO_2$ (>10,000 µatm) result in low seawater pH (<6) and the undersaturation of aragonite and calcite in an area devoid of calcified organisms such as shelled molluscs and hard corals. Around 100–400 m away from the Vulcano seeps, the geochemistry of the seawater becomes analogous to future ocean acidification conditions, with dissolved carbon dioxide levels falling from 900 to 420 µatm as seawater pH rises from 7.6 to 8.0 [74].

In this case study, the direct effect of $CO_2$ emissions on the sea bottom seems to be restricted to the neighbouring area of the carbonate conduits. In fact, evidence of acidification at the water–sediment interface is highlighted by the decrease in the abundance of microfauna (mainly benthic foraminifers) in sediments close to the conduits.

The indirect effect of $CO_2$ diffused (or trapped) in the clay sediments seems to be responsible for the acidification of interstitial (pore) water and the dissolution of

calcitic bivalve shells (i.e., *Megaxinus incrassatus*) recovered as internal moulds in the carbonate conduits.

$CO_2$ added to seawater changes the hydrogen ion concentration (pH); this may affect marine life through mechanisms that do not directly involve $CO_2$. At high $CO_2$ concentrations, animals can asphyxiate because their blood cannot transport enough oxygen to support metabolic functions. In the most active open ocean squid (*Illex ilecerebrosus*), model calculations predict acute lethal effects with a rise in $pCO_2$ of 6500 ppm and a 0.25 unit drop in blood pH. However, acute $CO_2$ sensitivity varies between squid species. It should be emphasised that squid are one of the main prey species of the sperm whale and some other toothed cetaceans [74].

However, direct effects of dissolved $CO_2$ on diving marine air breathers (mammals and turtles) can likely be excluded since they possess higher $pCO_2$ values in their body fluids than water breathers and gas exchange is minimised during diving. It is possible to speculate that marine mammals, such as sperm whales, may be indirectly affected through potential $CO_2$ effects on their preferred food, the squid.

*5.5. Geological and Palaeoenvironmental Implications*

Although no evidence of pervasive fracturing was noted in the field, observations from satellite images indicate the alignment of isolated structures throughout the riverbed, suggesting that their distribution is controlled by faulting (Figures 3b and 9). This alignment ranges between WNW and NW, and it is compatible with both the main trend of the basin (Figure 9) and the systems of fracture affecting the same clay in the Bargiano area [33,37,38]. This feature is well known and has been reported by several other researchers [2,75,76]. Although not conclusive, the presence of the study structure adds a new piece of evidence for reconstruction of the early Pleistocene tectonic and volcanic scenario. These vast fields of carbonate conduit concretions imply new considerations regarding the importance of hydrothermal $CO_2$ emissions during the Early Pleistocene in southwestern Umbria and evidence the existence of tectonic activity and volcanic phenomena during this interval of time (Figure 9). Moreover, both $CO_2$-related structures and whale-fall events (WFE) in the area are coeval with volcanic phase $V_1$ (Figure 2a; [28,34,59]); the presence of other, as yet undocumented, fields of mineralised structures in the area cannot be excluded.

The presence of this $CO_2$ degassing field and the emergence of volcanoclastic deposits [28,34,59] fits well with the local pattern of extensional faults (Figure 9). Such a situation is comparable with present-day $CO_2$ emissions and $CO_2$-rich springs [65] and is further evidence of the incipient volcanic activity of Mt. Vulsini since the Early Pleistocene. This phase was reasonably characterised by diffused, mainly submarine fissural activity, producing a dispersed pyroclastic fallout [59]. In this scenario, the occurrence of extensional faults also guided the rise of $CO_2$ to the surface and their trapment in clay sediments at shallow depths below the sea floor (Figure 11). In this tectonically active context, the presence of the described mineralised structures was related to the progressive onset of escape paths followed by hydrothermal/volcanic $CO_2$-rich fluids. This slow and progressive rise of fluids can justify both the almost exclusive presence of dolomite and the occurrence of thin calcite mineralisation only in the internal wall and in the mouths of conduits. According with [2], it is presumed that mineralisation took place at very shallow depths inside the sediment (i.e., just below the sea floor), with free gas emission at the water–sediment interface. The proposed model [2], linking mineralisation to fault systems, accounts well for the formation of large, multiple doughnut structures, and it is certainly applicable to this case study. Nonetheless, the relation with faulting seems to be here more generalised and related to a diffuse stress field rather than to main fault plains (i.e., the master faults bordering the basin, Figure 11). In fact, the accompanying volcanism progressively shifted southwestwards, during the Middle–Late Pleistocene (~750–100 ka), to the main area of the Vulsini Mts. (Figures 1 and 2).

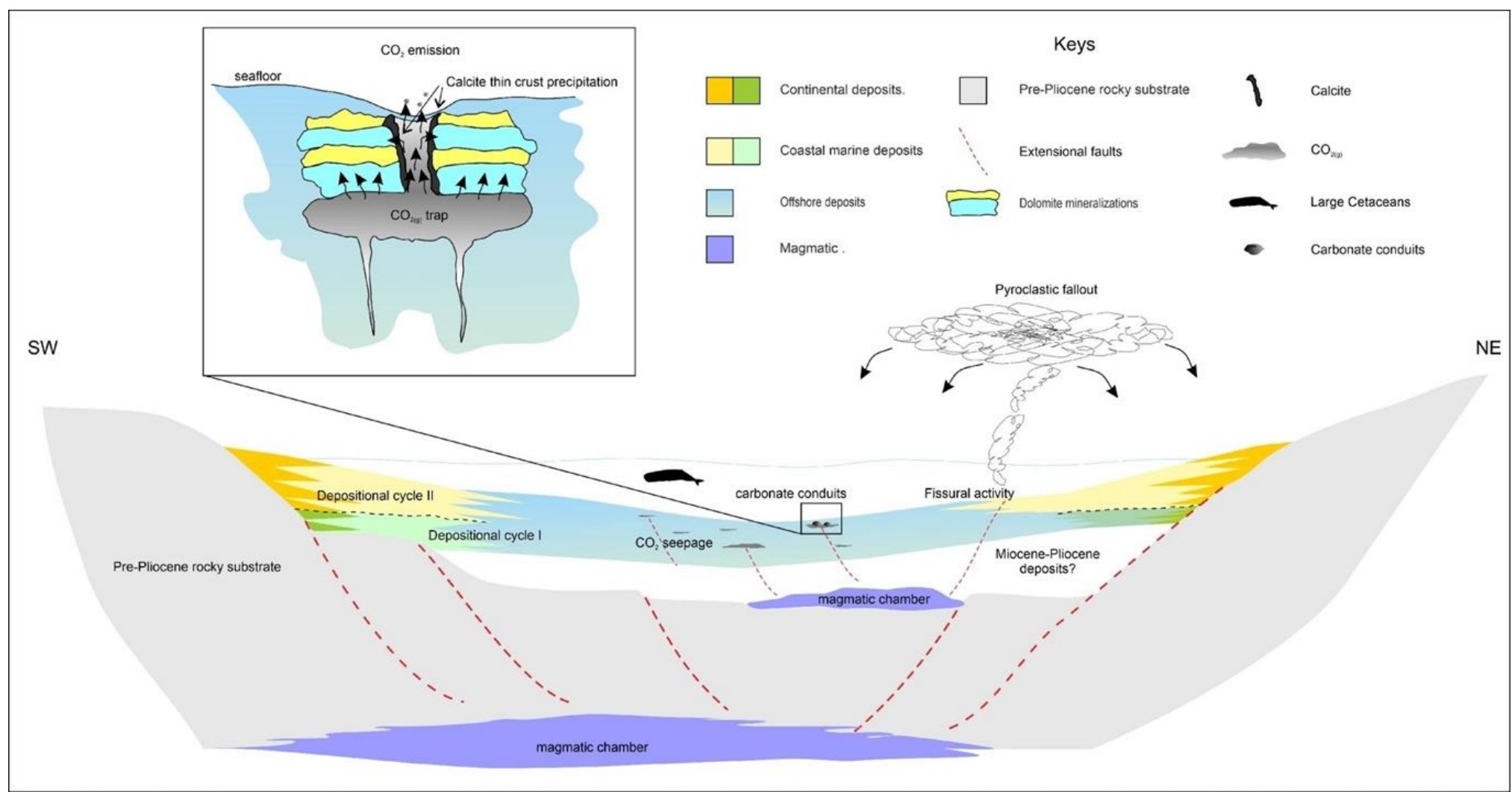

**Figure 11.** Proposed model for the origin of mineralised structures in relation to tectonic and volcanic phases.

The possible effects of $CO_2$ release in water on marine species have been discussed in Section 5.4. As shown in Figures 2 and 8, the spatial and temporal localisation of WFE is still compatible with the tectonic context, although a direct correlation with the presence of $CO_2$ in water is hardly justifiable. The high concentration of WFE in a restricted area still lacks a definitive interpretation [33,37,38]. The documentation of $CO_2$-degassing carbonate conduits not far from the sites of WFE adds another piece to the paleoenvironmental puzzle, particularly to the hypothesis that their presence was more widespread than that documented previously. In this way, the paleoenvironmental correlation, whilst still feasible, is more nuanced. At this stage, it is only possible to infer that tectonics and volcanism guided the onset of paleoenvironmental conditions favourable to the presence of great cetaceans and were indirectly responsible for mass-death events, acting on parameters that somehow influenced the food chain.

## 6. Conclusions

This study reports the first occurrence of carbonate conduits made of dolomite of abiogenic origin in the clay marine deposits of the early Pleistocene. The morphological features of the conduits, the positive isotopic values of carbon and oxygen in the carbonates, and the thermodynamic considerations regarding dolomite stability suggest that dolomite precipitation was related to the rise of hydrothermal/volcanic $CO_2$-rich fluids. This is in agreement with the early Pleistocene volcanic activity of the nearby Vulsini complex. The rising fluids were trapped inside the highly impermeable clay sediments until internal pressure reached the value of lithostatic pressure and broke the resistance of the hosting sediments, moving towards the seafloor. During their ascent, the hydrothermal/volcanic fluids mixed, at variable depths, with the seawater contained in the clay sediments, precipitating dolomite. The spatial distribution of the conduits clearly indicates that the fluid ascent followed preferential pathways corresponding to visible fractures related to early Pleistocene faults.

**Supplementary Materials:** The following supporting information can be downloaded at: https://www.mdpi.com/article/10.3390/min12070819/s1, Table S1: Gravimetric data from the cyclohexane (CH) and dichloromethane (DCM) extraction of carbonate conduit concretion samples and the extraction of esters following methyl esterification. Table S2: Calculated biomarker parameters. Figure S1: HTGC chromatograms of in-house $nC_{10–30,40,50,60}$ alkanes standard, and cyclohexane extracts from procedural blank and carbonate conduit concretion. Figure S2: HTGC-ToF-MS (12 eV) total ion chromatograms of the cyclohexane extracts of the procedural blank and carbonate conduit concretions. Figure S3: HTGC-ToF-MS (12 eV) extracted ion chromatograms (*m/z* 136) of the cyclohexane extracts of the procedural blank and carbonate conduit concretions. Figure S4a: HTGC-ToF-MS (12 eV) extracted ion chromatograms (*m/z* 476) of the cyclohexane extracts of the procedural blank and carbonate conduit concretions. Figure S4b: Zoomed region HTGC-ToF-MS (12 eV) extracted ion chromatograms (*m/z* 476) of the cyclohexane extracts of the procedural blank and carbonate conduit concretions. Figure S5a: 12 eV mass spectrum of peak at 10.81 min. Figure S5b: NIST library mass spectrum (70 eV) of squalene. Figure S6: Mass spectrum of peak I at 13.51 min. Figure S7: Mass spectrum of peak II at 13.70 min. Figure S8: Mass spectrum of peak at 13.89 min. Figure S9: Peak area ratios using *m/z* 476 extracted ions for peak II/peak I and for peak III/peak I. Figure S10: HTGC-MS extracted ion chromatograms (upper *m/z* 191; lower *m/z* 217) indicating distribution of triterpanes and sterane biomarkers in sample 2. Figure S11: HTGC chromatograms of in-house $nC_{10–30,40,50,60}n$-alkanes standard, and dichloromethane extracts of residues following cyclohexane extraction from procedural blank and carbonate conduit concretions. Figure S12: HTGC chromatograms of in-house $C_{8-20}n$-fatty acid methyl esters (*n*-FAMEs) standard, and cyclohexane (methyl ester) extracts of residues following cyclohexane and dichloromethane extraction from procedural blank and carbonate conduit concretions.

**Author Contributions:** Conceptualisation, A.B., R.B. and F.F.; methodology, A.B., R.B., C.B., F.F. (Francesco Frondini), M.L., S.R. and P.A.S.; formal analysis, C.B., F.F. (Francesco Frondini), M.L., S.R. and P.A.S.; investigation, A.B., R.B., C.B., F.F. (Francesco Frondini), F.F. (Federico Famiani), M.L., S.R. and P.A.S.; writing—original draft preparation, A.B., R.B., C.B., F.F. (Francesco Frondini), M.L., S.R. and P.A.S.; writing—review and editing, All authors; visualisation, A.B., R.B., C.B., F.F. (Francesco Frondini), M.L., S.R. and P.A.S.; funding acquisition, A.B. All authors have read and agreed to the published version of the manuscript.

**Funding:** This research was funded by A.B. research funds (RICVABALDA).

**Acknowledgments:** We want to thank Ermindo Tardiolo, a connoisseur of the territory of Allerona, who first pointed out to us the existence of these structures emerging along the bed of the Paglia River; without his contribution, we would never have been able to study this particular topic. We also want to thank the three anonymous reviewers who contributed to the improvement of the initial manuscript.

**Conflicts of Interest:** The authors declare no conflict of interest.

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
