# Peer review of "CO2-Degassing Carbonate Conduits in Early Pleistocene Marine Clayey Deposits in Southwestern Umbria (Central Italy)"

_minerals, doi:10.3390/min12070819_

Round 1

Reviewer 1 Report

The manuscript: minerals-1695170 titled “CO2 degassing carbonate conduits in Early Pleistocene marine clayey deposits from Southwestern Umbria (Central Italy)”, presents an interesting research study. The manuscript is well written, with clear aims set in the introduction, and in addition Figures are correctly displayed. Results are given in a scientifically proper manner providing the necessary support for the discussion section which is quite informative. Conclusions and Abstract are balanced, well written, and successfully display the major outlines of this paper. Therefore, I suggest that this paper should be published in its present form in the Journal of “Minerals”.

Author Response

We thank you for your favorable opinion and for your interest in our research.

the authors

Reviewer 2 Report

I read with interest the manuscript “CO2 degassing carbonate conduits in Early Pleistocene marine clayey deposits from Southwestern Umbria (Central Italy)” (Special Issue: “Hydrothermal Systems Across Time and Space: Advances and Perspectives”). The authors investigate several carbonate conduits in Early Pleistocene marine clayey deposits combining mineralogical, stable isotopes and organic geochemical analyses. They mainly focus on understanding the origin of the conduits, suggesting that it is driven by the rise of hydrothermal-volcanic CO2-rich fluids through extensional fault systems.

General comments

Overall, I find this manuscript interesting and well-structured. A detailed work has been done, providing a coherent dataset through numerous and accurate analyses. However, there are a number of aspects that require further attention.

I think that the authors should provide a much more detailed discussion on their hypothesis of a hydrothermal-volcanic origin of CO2-rich fluids. Particularly, stable isotopes of the carbonate samples, especially δ13C, are mainly used in the manuscript to quantitatively constrain the source of the precipitated carbonate. However, a direct emission of hydrothermal-volcanic CO2 is just presented as an alternative non-biological origin, and lacks supporting data, calculations and references. For example, the measured, overall positive δ13C is linked to hydrothermal-volcanic origin in the paper, however several available literature data are inconsistent with this. E.g., Simmons and Christenson (1994, Am. J. Sci.) at Broadlands-Ohaaki (Taupo District, New Zeland) and Chiodini et al (2015; Geology) at Campi Flegrei caldera (Italy) found stable, negative δ13C (representative of magmatic influx) for hydrothermal carbonate minerals precipitated in volcanic areas, down to low formation temperatures. I do not exclude a hydrothermal-volcanic origin but more quantitative constraints are needed as it is a key point of the manuscript (and would be an interesting conclusion to better understand past volcanism in this area).

Moreover, I think that final aims of the paper and reasons that motivated the study of these structures should be better highlighted and linked with the achieved results. The paper explores specific structures in a localized area. Much attention in the introduction, as well as in the conclusions and abstract, is just paid to specific features and origin of these structures. Different kinds of implications are only presented in the final part of the discussion section. In my opinion, the authors should better contextualize the investigation of these conduits and clarify why it can be relevant.    

Specific comments

Lines 19-20: I suggest to provide first the results and then information on data interpretation

Lines 29-30: It can be useful to highlight that it is an incipient volcanic activity of Mt. Vulsini (as volcanism at Mt. Vulsini is usually referred to about 0.6-0.15 Ma, e.g., Peccerillo, 2005, Springer; Peccerillo and Frezzotti, 2015, JGS)

Lines 48-49: Please change “it is thought” with “we thought” or “At the moment” to “To our knowledge” or similar

Line 63: Remove strikethrough after square bracket

Lines 78-82: Please clarify the sentences of the paragraph. E.g., What “they” (line 79) are? Close the parenthesis in line 80

Line 89: Add interval name as in Figure 2a

Line 93: Remove strikethrough for “Mts”

Figure 1: If it is possible, move coordinate numbers outside the map

Lines 145-146: More information on the sampling, and especially on sampling location, is needed. From what heights were they taken? Do they come from centers or peripheries?

Lines 183-184: An overview on morphological data (e.g., min and max diameters and heights of the different types of conduits) can be useful

Line 186 and Table 1: Maybe “troncoconical” is more correct

Line 215: Add structure number as in line 213

Sections 4.2 to 4.5: It can be useful to describe how/if the data changes as the sample location changes (e.g., moving from top to down or from centers to peripheries in a conduit, as well as considering the coordinates of the conduits)

Line 287-288: Specify sample names in brackets

Lines 338-342: Also the hypothesis of residual CO2 from methanogenesis can lead to positive carbon signatures and should be discussed here (e.g., see Nelson et al., 2017; New Zealand J. Geol. Geophys.)

Line 341: As suggested in the general comments, the hypothesis of a hydrothermal-volcanic origin of CO2-rich fluids should be quantitatively constrained with more data, calculations and references. Moreover, I’m not sure that the cited paper of Aiuppa et al. discusses carbon isotopes and is properly cited

Lines 346-347: Please specify and detail the data that was used for this calculation

Author Response

Reply to  Reviewer #2

General comments

Overall, I find this manuscript interesting and well-structured. A detailed work has been done, providing a coherent dataset through numerous and accurate analyses. However, there are a number of aspects that require further attention.

I think that the authors should provide a much more detailed discussion on their hypothesis of a hydrothermal-volcanic origin of CO2-rich fluids. Particularly, stable isotopes of the carbonate samples, especially δ13C, are mainly used in the manuscript to quantitatively constrain the source of the precipitated carbonate. However, a direct emission of hydrothermal-volcanic CO2 is just presented as an alternative non-biological origin, and lacks supporting data, calculations and references. For example, the measured, overall positive δ13C is linked to hydrothermal-volcanic origin in the paper, however several available literature data are inconsistent with this. E.g., Simmons and Christenson (1994, Am. J. Sci.) at Broadlands-Ohaaki (Taupo District, New Zeland) and Chiodini et al (2015; Geology) at Campi Flegrei caldera (Italy) found stable, negative δ13C (representative of magmatic influx) for hydrothermal carbonate minerals precipitated in volcanic areas, down to low formation temperatures. I do not exclude a hydrothermal-volcanic origin but more quantitative constraints are needed as it is a key point of the manuscript (and would be an interesting conclusion to better understand past volcanism in this area).

Reply: In the text, we clearly show how starting from a “volcanic” isotopic composition of the CO2 it is possible, through fractionation, to get the isotopic composition of the dolomite. This part is reported in 5.1 paragraph.

Moreover, I think that final aims of the paper and reasons that motivated the study of these structures should be better highlighted and linked with the achieved results. The paper explores specific structures in a localized area. Much attention in the introduction, as well as in the conclusions and abstract, is just paid to specific features and origin of these structures. Different kinds of implications are only presented in the final part of the discussion section. In my opinion, the authors should better contextualize the investigation of these conduits and clarify why it can be relevant.    

Reply: thanks for the suggestions, we modified the text.

 Specific comments

Lines 19-20: I suggest to provide first the results and then information on data interpretation

Reply: modified as suggested

Lines 29-30: It can be useful to highlight that it is an incipient volcanic activity of Mt. Vulsini (as volcanism at Mt. Vulsini is usually referred to about 0.6-0.15 Ma, e.g., Peccerillo, 2005, Springer; Peccerillo and Frezzotti, 2015, JGS)

Reply: It is right the  activity of Mts. Vulsini is usually constrained in the range indicated, but also consider an incipient volcanism still related to Vulsini has been documented (e.g., Petrelli et al. 2017, Bizzarri et al. 2011, Bizzarri and Baldanza 2020). In this paper, we consider the study structures as related to these older phenomena. As citations are not allowed in the abstract, this cannot be pointed out here. Anyway, we added the reference to Peccerillo and Frezzotti 2015 in the geological setting section, and we recalled it also in the discussions. 

Lines 48-49: Please change “it is thought” with “we thought” or “At the moment” to “To our knowledge” or similar

Reply: modified as suggested

Line 63: Remove strikethrough after square bracket

Reply: modified as required

Lines 78-82: Please clarify the sentences of the paragraph. E.g., What “they” (line 79) are? Close the parenthesis in line 80

Reply: the sentence has been modified

Line 89: Add interval name as in Figure 2a

Reply: interval names have been indicated accordingly to Figure 2a

Line 93: Remove strikethrough for “Mts”

Reply: modified as required

Figure 1: If it is possible, move coordinate numbers outside the map

Reply: Figure 1 has been modified as required

Lines 145-146: More information on the sampling, and especially on sampling location, is needed. From what heights were they taken? Do they come from centers or peripheries?

Reply: Please consider the total, real vertical extension of the structures was not directly visible in the outcrop, nor otherwise reconstructable. The structures cropped out for limited eight from the Paglia River bed, thus samples were taken from the outcropping, accessible and very superficial portion. To provide the information required, we added sampling points directly in Figure 4 and a dedicated Table (the new Table 2).

Lines 183-184: An overview on morphological data (e.g., min and max diameters and heights of the different types of conduits) can be useful

Reply: We added a few lines about this point in the results.

Line 186 and Table 1: Maybe “troncoconical” is more correct

Reply: We substitute “tronchoconical” with troncoconical

Line 215: Add structure number as in line 213

Reply: Modified as required.

Sections 4.2 to 4.5: It can be useful to describe how/if the data changes as the sample location changes (e.g., moving from top to down or from centers to peripheries in a conduit, as well as considering the coordinates of the conduits)

Reply: No are evidences of changes in mineralogical and petrografic features. The only difference concerns the two type of matrix (see the description in 4.2 paraghraph).

Line 287-288: Specify sample names in brackets

Reply: Modified as required

Lines 338-342: Also the hypothesis of residual CO2 from methanogenesis can lead to positive carbon signatures and should be discussed here (e.g., see Nelson et al., 2017; New Zealand J. Geol. Geophys.)

Reply: A specific part of comparison with the data of Nelson et al 2017 has been added.

Line 341: As suggested in the general comments, the hypothesis of a hydrothermal-volcanic origin of CO2-rich fluids should be quantitatively constrained with more data, calculations and references. Moreover, I’m not sure that the cited paper of Aiuppa et al. discusses carbon isotopes and is properly cited

Reply: The citation of Aiuppa et al. was incorrectly positioned in the text, sorry.

Lines 346-347: Please specify and detail the data that was used for this calculation

Reply: Specified as required

Reviewer 3 Report

The origin and structure of fluid conduits have been important topics for both the tectonic and metallogenic communities. This manuscript presents field observations along with mineralogic, isotopic, and biochemical data of the fluid conduits in early Pleistocene marine clayey deposits, southwestern Umbria. The authors suggest that (1) The carbonate conduits were constructed by non-biological carbonate rocks that were probably related to volcanic CO2 emission; (2) The migration of the CO2 bearing fluid is controlled by regional faults, which is evidenced by the spatial distribution of the conduits; (3) The fluids were trapped and cumulated in the clayey sediment. Once the fluid pressure exceeds the country rock’s tensile strength, hydraulic fracturing and mineral precipitation would contribute to the development of the conduits.

By and large, this study provides valuable documentation on the structure and evolution of carbonate fluid conduits in the study area. Principal conclusions are generally supported by available data. However, revisions are needed before the manuscript can be published in MINERALS. Detailed comments are given below.

(1) The manuscript provides ample pieces of evidence in support of the origin of the CO2-rich fluid. However, another major argument, the structural controls on the fluid conduits, is less discussed;

(2) The introduction section should be revised to emphasize the significance of this work. According to the current version, the authors argued that the studies on the fluid conduits related to CO2 emission are relatively rare. However, this could not explain the necessity of this study.

(3) Some of the statements are confusing. For example, in lines 30-31, “Vertical and horizontal developments of the structures suggest a possible rise of carbon dioxide pockets, buried inside the clayey sediments, along a fault system”. Maybe the authors are trying to state that (1) the upward migration of the CO2 was controlled by the fault system; (2) The carbon dioxide pockets were buried and accumulated in the clayey sediments until the pressure exceed the eruptive threshold.

(4) Extensive English editing is needed to make the manuscript more comprehensive and easier to follow. For instance, in line 29, “marine Early Pleistocene deposits” could be replaced by “Early Pleistocene marine deposits”

Specific comments:

Line 93 “Vulsini Mts, Vico, and Sabatini Mts.” should be “Vulsini, Vico, and Sabatini Mts.”.

Author Response

Reply to  Reviewer # 3

The origin and structure of fluid conduits have been important topics for both the tectonic and metallogenic communities. This manuscript presents field observations along with mineralogic, isotopic, and biochemical data of the fluid conduits in early Pleistocene marine clayey deposits, southwestern Umbria. The authors suggest that (1) The carbonate conduits were constructed by non-biological carbonate rocks that were probably related to volcanic CO2 emission; (2) The migration of the CO2 bearing fluid is controlled by regional faults, which is evidenced by the spatial distribution of the conduits; (3) The fluids were trapped and cumulated in the clayey sediment. Once the fluid pressure exceeds the country rock’s tensile strength, hydraulic fracturing and mineral precipitation would contribute to the development of the conduits.

By and large, this study provides valuable documentation on the structure and evolution of carbonate fluid conduits in the study area. Principal conclusions are generally supported by available data. However, revisions are needed before the manuscript can be published in MINERALS. Detailed comments are given below.

 (1) The manuscript provides ample pieces of evidence in support of the origin of the CO2-rich fluid. However, another major argument, the structural controls on the fluid conduits, is less discussed;

Reply: A few lines have been added in discussions. Nonetheless, please consider all data (alignment of structures, evidences of coeval volcanism, whale -fall events) rather than a local pattern of fractures point to a complex of tectonic-driven setting.

Reply: we modified according to the suggestions.

 (3) Some of the statements are confusing. For example, in lines 30-31, “Vertical and horizontal developments of the structures suggest a possible rise of carbon dioxide pockets, buried inside the clayey sediments, along a fault system”. Maybe the authors are trying to state that (1) the upward migration of the CO2 was controlled by the fault system; (2) The carbon dioxide pockets were buried and accumulated in the clayey sediments until the pressure exceed the eruptive threshold.

Reply: we modified the text following the suggestions.

 (4) Extensive English editing is needed to make the manuscript more comprehensive and easier to follow. For instance, in line 29, “marine Early Pleistocene deposits” could be replaced by “Early Pleistocene marine deposits”

Reply: the final English check has been done by two co-authors, native English speakers

 Specific comments:

 Line 93 “Vulsini Mts, Vico, and Sabatini Mts.” should be “Vulsini, Vico, and Sabatini Mts.”.

Reply: modified

Round 2

Reviewer 2 Report

In my opinion the authors have properly revised the manuscript based on the reviewers' comments. The revised manuscript is sufficiently improved and I suggest to accept the present version.

Just two final suggestions for the section 5.1:

- the authors could mention/discuss that a more positive carbon isotopic composition can be also related to fractionation at the low estimated temperatures (e.g., see Bottinga et al., 1968,  J. Phys. Chem., for calcite)

- it can be useful to add the value used for present seawater d18O, and if/how fractionation mechanisms can impact on the calculated temperature, in the line 392.

Author Response

Dear Reviewer,

Regarding the two final suggestions for the section 5.1:

- the authors could mention/discuss that a more positive carbon isotopic composition can be also related to fractionation at the low estimated temperatures (e.g., see Bottinga et al., 1968, J. Phys. Chem., for calcite)

 Reply: We add the low temperature at line 379

- it can be useful to add the value used for present seawater d18O, and if/how fractionation mechanisms can impact on the calculated temperature, in the line 392.

 Reply: We add the value of seawater d18O

Thanks for helping to improve the text.

Best regards